# MORE SIDE INFORMATION, BETTER PRUNING: SHARED-LABEL CLASSIFICATION AS A CASE STUDY

## ABSTRACT

Pruning of neural networks, also known as compression or sparsification, is the task of converting a given network, which may be too expensive to use (in prediction) on low resource platforms, with another 'lean' network which performs almost as well as the original one, while using considerably fewer resources. By turning the compression ratio knob, the practitioner can trade off the information gain versus the necessary computational resources, where information gain is a measure of reduction of uncertainty in the prediction.

In certain cases, however, the practitioner may readily possess some information on the prediction from other sources. The main question we study here is, whether it is possible to take advantage of the additional side information, in order to further reduce the computational resources, in tandem with the pruning process?

Motivated by a real-world application, we distill the following elegantly stated problem. We are given a multi-class prediction problem, combined with a (possibly pre-trained) network architecture for solving it on a given instance distribution, and also a method for pruning the network to allow trading off prediction speed with accuracy. We assume the network and the pruning methods are state-of-the-art, and it is not our goal here to improve them. However, instead of being asked to predict a single drawn instance $x$, we are being asked to predict the label of an $n$-tuple of instances $(x_1, \ldots x_n)$, with the additional side information of *all tuple instances share the same label*. The shared label distribution is identical to the distribution on which the network was trained.

One trivial way to do this is by obtaining individual raw predictions for each of the $n$ instances (separately), using our given network, pruned for a desired accuracy, then taking the average to obtain a single more accurate prediction. This is simple to implement but intuitively sub-optimal, because the $n$ independent instantiations of the network do not share any information, and would probably waste resources on overlapping computation.

We propose various methods for performing this task, and compare them using extensive experiments on public benchmark data sets for image classification. Our comparison is based on measures of relative information (RI) and $n$-accuracy, which we define. Interestingly, we empirically find that i) sharing information between the $n$ independently computed hidden representations of $x_1, .., x_n$, using an LSTM based gadget, performs best, among all methods we experiment with, ii) for all methods studied, we exhibit a *sweet spot* phenomenon, which sheds light on the compression-information trade-off and may assist a practitioner to choose the desired compression ratio.

## 1 INTRODUCTION

Pruning Neural networks, the task of compressing a network by removing parameters, has been an important subject both for practical deployment and theoretical research. Some pruning algorithms have focused on manipulating pre-trained models, (Mozer & Smolensky, 1989; LeCun et al., 1990; Reed, 1993; Han et al., 2015) while recent work have identified that there exist sparse subnetwork (also called winning tickets) in randomly-initialized neural networks that, when trained in isolation, can match and often even surpass the test accuracy of the original network (Frankle & Carbin, 2019;

Frankle et al., 2020). There is a vast literature on network pruning, and we refer the reader to Blalock et al. (2020); Sze et al. (2017); Reed (1993) for an excellent survey. In this work, we adopt the pruning methods of Tanaka et al. (2020); Lee et al. (2019); Wang et al. (2020); Han et al. (2015) which have been influential in our experiments.

More crucially, most literature on pruning has been focused on designing a machine that converts a fixed deep learning solution to a prediction problem, to a more efficient version thereof. The pruning machine has a *compression knob* which trades off the level of pruning with accuracy of the prediction. The more resources we are willing to expend in prediction (measured here using floating-point operations (FLOPs)), the more information we can obtain, where information here is measured as prediction accuracy, or as reduction of uncertainty (defined below).

We now ask what happens when we want to prune a network, but also possess information on the prediction coming from another source. Intuitively, given some form of additional side information, we should be able to prune our network with a higher compression ratio to reach the same level of accuracy for the prediction task, compared with a scenario with no additional side information. But how can we take the side information into account when pruning?

## 1.1 MOTIVATION

This question was motivated by an actual real-life scenario. We describe the scenario in detail, although the actual problem we thoroughly study in what follows is much simpler.

Imagine a database retrieval system with a static space of objects $\mathcal{X}$. Given a query object $q$, the goal is to return an object $x$ from $\mathcal{X}$ that maximizes a ground-truth retrieval value function $f_q(x)$. We have access to a function $\tilde{f}_q(x)$ expressed as a deep network, which approximates $f_q$, and was trained using samples thereof. The function $\tilde{f}_q$ is very expensive to compute. (Note that we keep $q$ fixed here, as part of the definition of $f_q(\cdot)$, although in an actual setting both $q$ and $x$ would be input to a bivariate retrieval function $\tilde{f}$.) Computing $\tilde{f}_q(x)$ for all $x \in \mathcal{X}$ is infeasible. One way to circumvent this is by computing a less accurate, but efficient function $\tilde{\tilde{f}}_q(\cdot)$, defined by the network resulting in a pruning of the network defining $\tilde{f}_q$. Then compute $\tilde{\tilde{f}}_q(\cdot)$ on all $x \in \mathcal{X}$ to obtain a shortlist of candidates $\mathcal{X}'$, and then compute $\tilde{f}_q(x)$ on $x \in \mathcal{X}'$ only. This idea can also be bootstrapped, using rougher, more aggresively pruned estimates $\tilde{\tilde{\tilde{f}}}_q, \tilde{f}_q^{(4)}, \tilde{f}_q^{(5)}...$ and increasingly shorter shortlist. However, an important point is ignored in this approach: The space $\mathcal{X}$ is structured, and we expect there to be prior connections between its elements. This is the *side information*. Such connections can be encoded, for example, as a similarity graph over $\mathcal{X}$ where it is expected that $f_q(x_1)$ is close to $f_q(x_2)$ whenever there is an edge between $x_1, x_2$. There is much work on deep networks over graphs (Zhou et al., 2018; Kipf & Welling, 2017; Wu et al., 2020). But how can the extra information, encoded as a graph, be used in conjunction with the pruning process?

Let us simplify the information retrieval scenario. First, assume that we are in a classification and not in a regression scenario, so that $f_q(x)$ can take a finite set of discrete values, and $\tilde{f}_q(x)$ returns a vector of logits, one coordinate per class. Second, assume the side information on $\mathcal{X}$ is a partitioning of $\mathcal{X}$ into cliques, or clusters $\mathcal{X}_1...\mathcal{X}_k$ where on each clique the value of $f_q(\cdot)$ is *fixed*, and written as $f_q(\mathcal{X}_i), i = 1..k$. Now the problem becomes that of estimating the $f_q(\mathcal{X}_i)$'s using $n$ random samples $x_{i1}...x_{in} \in \mathcal{X}_i, i = 1..k$. [1]

Fixing the cluster $\mathcal{X}_i$, one obvious thing to do in order to estimate $f_q(\mathcal{X}_i)$ is to take an average of the logit vectors $\tilde{f}_q(x_{i1})...\tilde{f}_q(x_{in})$, where $\tilde{f}_q$ is some fixed (possibly pruned) network, and use the $argmax$ coordinate as prediction. Assuming each pruned network $\tilde{f}_q$ outputs a prediction vector with a certain level of uncertainty, the averaged vector should have lower uncertainty, and this can be quantified using simple probabilistic arguments. This will henceforth be called the *baseline* method. Intuitively the baseline method, though easy to do using out-of-the-box pruning libraries, cannot possibly be optimal given the side information of *same label across $\mathcal{X}_i$*. Indeed, the baseline method feeds all the examples $x_{i1}...x_{in}$ independently through separate instantiations of $\tilde{f}_q$, and nothing

---

[1]Continuing the retrieval story , the practitioner would now find the $\mathcal{X}_i$ that maximizes $f_q$, and then further focus the search in that cluster.

prevents the different instantiations to learn overlapping pieces of information. Hence it makes sense to somehow interconnect these networks as a meta-network, and possibly do the pruning on the meta-network. In this work, we experiment with several methods for performing this task, and compare our results with the baseline.

## 1.2 THE SHARED-LABEL PREDICTION PROBLEM

We depart from the original motivating information retrieval scenario, and henceforth consider a simpler, toy problem which we call the *shared-label* prediction problem. We are given an underlying space of instances $\mathcal{X}$ and an unknown ground truth labelling function $f : \mathcal{X} \mapsto \mathcal{Y}$ for some discrete set $\mathcal{Y}$ of labels. The goal is to train a classifier that, given a random $n$-tuple of instances $x_1...x_n \in \mathcal{X}^n$ sharing the same unknown label $y$ (so that $f(x_1) = \cdots = f(x_m) = y$), outputs a prediction of $y$. This is the *shared prediction* problem.

Our work is empirical, and the goal is to develop general methods for the shared prediction problem, given a base network, designed for the standard (non-shared) prediction problem, and a base pruning method, we ask: *How do we reuse and rewire these readily available tools to effectively solve the shared-label prediction problem on tuples of $n$-instances?*

## 2 OUR CONTRIBUTION

Below in Section 2.1 we present four methods. Each method uses a baseline CNN model, together with a pruning method with a compression ratio knob $\rho$, and creates a meta-network that is parameterized by the information size $n$ and by $\rho$, designed to solve the shared classification problem. To measure our success, we will both use a measure of accuracy as well as a measure of relative information which we define below. We will compute these measures extensively over a grid of possible pairs $(n, \rho)$, for each method. Visualization of the results highlights an interesting invariant that is worth studying.

Intuitively, the measure of relative information tells us how efficiently each method uses its computational resource, without wasting time on computing the same pieces of information over and over on the $n$-tuple of instances. Therefore, it allows us to obtain a quantitative comparison between the methods. To define the measure, we first recall some information theory.

Given a random variable $Y$ over a discrete space, the Shannon entropy, or uncertainty of $Y$ is $H[Y] = -\sum \Pr[Y = y] \log \Pr[Y = y]$, where the sum ranges over possible values of $Y$. In our case, we will use $H(Y)$ to measure the uncertainty in the label of a randomly drawn instance, which is also the uncertainty in the label of a randomly drawn $n$-tuple in the shared label setting.

Given a random variable $\tilde{Y}$ (an estimate of $Y$), the information gain measures the difference between the entropy of $Y$ and the expectation with respect to $\tilde{Y}$ of $H(Y|\tilde{Y})$. More precisely, $IG(Y;\tilde{Y}) = H(Y) - E_{\tilde{Y}}\left[ -\sum_y \Pr[Y = y|\tilde{Y}] \log \Pr[Y = y|\tilde{Y}] \right]$. Note that information gain is symmetrical, that is $IG(\tilde{Y};Y) = IG(Y;\tilde{Y})$. Therefore it is also called *mutual information* and denoted $I(\tilde{Y};Y)$. In our setting, $\tilde{Y}$ will be an estimator of $Y$ obtained using the output of the network on an $n$-tuple of instances in the shared label setting, and $I(\tilde{Y};Y)$ will measure the expected amount of information we learn about $Y$ using the network output on that tuple. For a given network, we will be computing $IG(\tilde{Y};Y)$ empirically in what follows, by taking $\tilde{Y}$ to be the prediction obtained by selecting the argmax coordinate (logit) of the output of a network.

Given a method for the shared-label scenario, we define the relative information (RI) to be

$$\mathrm{RI}(\tilde{\mathrm{Y}}, \mathrm{Y}, \mathrm{n}, \rho) = \frac{IG(\tilde{Y};Y)}{n/\rho} \ .$$

In words, this is a measure of information that the network learns, per computational cost. The denominator $n/\rho$ is a reasonable measure of computational cost for the methods we study, because for these methods, the amount of computational effort we expend for shared label instance $x_1...x_n$

is proportional to $n$, and inverse proportional to the compression ratio $\rho$. We believe it is also a reasonable measure of computational cost for other natural methods.

For all methods we study, fixing the information size $n$, our experiments suggest that there exists a *sweet spot* phenomenon, or a "compression threshold" in the sense that RI, as a function of $\rho$, has a global maximum $\rho^*$. If the compression ratio $\rho$ is smaller than $\rho^*$, than we are at the *under-compressed regime*, where we can still save computational resources without **relatively** deteriorating the results, or the information, to a large extent. On the other side, if the compression ratio $\rho$ is bigger than $\rho^*$, than we are at the *over-compressed regime*, where we can gain a lot more information, by using a **relatively** mere amount of computational resources. We believe that a better understanding of this phenomenon can shed light on the interaction between different compression ratios, information sizes, and the information gains achieved by the methods (which is equivalent to test performances, as our experiments show). We show that the above is a robust phenomenon that occurs in a variety of settings.

## 2.1 OUR METHODS

1. *Baseline* method (Section 4.1) - Use a fixed model, with a fixed pruning method. For prediction, run the pruned model on the $n$ instances $x_1...x_n$, and use the average of the corresponding logit vectors for the shared prediction.

2. *Balanced* method (Section 4.2) - The same as the baseline method, except that the training set is organized such that each batch of images consists of $k$ random $n$-tuples, such that instances of each $n$-tuple share the same unknown label $y$. The model is trained using the *balanced* loss, which is a convex combination of a loss defined for $n$-tuples, and the standard loss on the individual instances.

3. *Graph* method (Section 4.3) - Inspired by work on Graph neural networks (GNNs), we propose an architecture consisting of $n$ duplicates of a base CNN, with information passage between neurons of the different copies of the CNN. The training set is organized in the same way as in the balanced method.

4. *Unified CNN-LSTM* method (Section 4.4) - We propose a model that combines a truncated version of a base CNN, giving a latent representation of the inputs, and then connecting the $n$ representations to each other, sequentially, using LSTM (Long Short-Term Memory) gadgets. Intuitively, this architecture uses information learned from instances $x_1...x_{i-1}$, encoded inside the LSTM, to assist in predicting $x_i$ for $i = 2, \ldots n$. The training set is organized in the same way as in the balanced method.

In Section 5.1 we validate the above *sweet spot* phenomenon under a variety of benchmark datasets, architectures, compression ratios and information lengths. We report the results of the baseline method for its simplicity (the same results hold for all other shared-label prediction methods as well).

In Section 5.2 we compare the differences between the baseline methods and the balanced method both qualitatively and quantitatively.

In Section 5.3 we compare all the proposed methods for shared-label prediction across different benchmark data sets for image classification and different evaluation metrics. The proposed unified CNN-LSTM method achieves significantly better performance compared to the other methods.

## 3 RELATED WORK

There is a variety of approaches to compressing neural networks, such as neural network pruning (Mozer & Smolensky, 1989; LeCun et al., 1990; Sze et al., 2017; Reed, 1993; Han et al., 2015; Blalock et al., 2020; Frankle & Carbin, 2019; Frankle et al., 2020), training of dynamic sparse networks (Bellec et al., 2018; Mocanu et al., 2018) dimensionality reduction of network parameters (Jaderberg et al., 2014; Novikov et al., 2015), and many more. Nonetheless, these results do not mention how the new compressed, efficient network, benefit from additional side information.

Moreover, there is much work on the "double-descent" phenomenon (Belkin et al., 2019; Advani & Saxe, 2017; Geiger et al., 2019). In a work by Nakkiran et al. (2020), it is shown that a variety of modern deep learning tasks exhibit a "double-descent", and that it occurs not just as a function

of model size. Therefore, it is an interesting question to ask whether this also occurs in the case of relative information, and our experimental results validate that this is not the case.

The concept of graph neural network (GNN) was first proposed by Scarselli et al. (2009), who extended existing neural networks for processing the data represented in graph domains graph papers. The first motivation of GNNs roots in convolutional neural networks (CNNs) (Lecun et al., 1998). Recent works on GNNs (Zhou et al., 2018; Kipf & Welling, 2017; Wu et al., 2020) inspired us to extend this idea as one of our methods for the task of shared-label prediction.

Lastly, RNNs are interesting for our purposes because they equip neural networks with memory, and the introduction of gating units such as LSTM and GRU (Hochreiter & Schmidhuber, 1997; Cho et al., 2014) has greatly helped in making the learning of these networks manageable. The LSTM based architecture has yielded the most promising results throughout our experiments.

## 4 OUR METHODS

In order to describe our four methods in details, we will need to present some standard terminology from the network pruning literature.

*Layer-collapse* - Pruning neural networks is usually done in two steps: The first step scores the parameters of a network according to some metric and the second step eliminates parameters based on their scores. This process can be applied both globally (on the network as a whole) and locally (separately on each layer). Recent work (Wang et al., 2020; Lee et al., 2020; You et al., 2020) has identified a key failure mode, layer-collapse, for the global version. Layer-collapse occurs when an algorithm prunes all parameters in a layer, rendering the network disconnected (and untrainable).

*Compression ratio ($\rho$)* - Logarithm to the base 10 of the number of parameters in the original network divided by the number of parameters remaining after pruning. In our experiments we use (not necessarily integer) powers of 10 for the compression ratios. For example, compression = 2.5 means that the number of parameters in the original network divided by the number of parameters remaining after pruning equals to $10^{2.5}$

*Max compression ($\rho_{max}$)* - The maximal possible compression ratio for a network that doesn't lead to layer-collapse.

We further define an accuracy-based evaluation metric for a shared-label prediction method, *n-accuracy*, which highly correlates with information gain, as our experiments show. We denote $\ell$ to be the number of classes in the data set and without loss of generality let the labels be $\{1, ..., \ell\}$. Moreover, $\tilde{y}_i$ is a vector of size $\ell$ that contains raw, unnormalized scores for each class, predicted by a given model.

*n-accuracy* - the percentage of correctly classified $n$-tuples. Formally, $n$-accuracy of a shared-label prediction method is defined to be

$$\frac{1}{T} \sum_{i=1}^{T} X_i$$

where $X_i$ is an indicator for the event that the $i$'th $n$-tuple was classified correctly. In our experiments we take $T = \ell \cdot 100$. Namely, we test on 100 random $n$-tuples from each class and report the average accuracy.

### 4.1 BASELINE METHOD

In the baseline method, each model is simply trained in a standard fashion for image classification, with a randomly shuffled training set with batch size $B$ and the Cross Entropy Loss defined as:

$$Cross\ entropy\ Loss(\tilde{y}, class) = -log(\frac{exp(\tilde{y}[class])}{\sum_j exp(\tilde{y}[j])}) = -\tilde{y}[class] + log(\sum_j exp(\tilde{y}[j]))$$

Finally, the losses are averaged over observations for each batch:

$$Standard\ Loss(batch) = \frac{1}{B} \cdot \sum_{i=1}^{B} Cross\ entropy\ Loss(\tilde{y}_i, class_i)$$

where $\tilde{y}_i$ contains raw, unnormalized scores for each class, predicted by the model for the $i$th data point in the batch and $class_i$ is its corresponding label. For this method, evaluation is done by simply taking an average of the predicted logit vectors $\tilde{y}_1...\tilde{y}_n$, and then taking the $argmax$ as the shared label prediction. In this way, we can take advantage of the probability scores in each logit vector.

## 4.2 BALANCED METHOD

Recall that our task is to classify $n$ different data points that share the same class with their corresponding label. Thus, the motivation will be to optimize directly for that purpose. The training set is organized such that each batch of images of size $B$ consists $k$ $n$-tuples, such that each $n$-tuple share the same unknown label $y$ ($B = k \cdot n$). For a batch of size $B$, the average batch prediction is defined to be:

$$Average\ Batch\ Prediction(batch) = \frac{1}{B} \cdot \sum_{i=1}^{B} \tilde{y}_i$$

Furthermore, let $batch_i$ be the subset of the current batch that only contains data points corresponding to label $i$ ($batch_i$ is of size $k$). Denote $\bar{y}^i \equiv Average\ Batch\ Prediction(batch_i)$. Then, the Average Same Label Loss is defined to be:

$$Average\ Same\ Label\ Loss(batch) = \frac{1}{\ell} \cdot \sum_{j=i}^{\ell} Cross\ entropy\ Loss(\bar{y}^i, i)$$

Intuitively, the loss function encourages the model to do well on each $n$-tuple rather than doing well on each specific data point. As a result, using this loss as it is in our experiments, does not lead to a model that generalizes well. Therefore, we offer a natural trade-off between the *Standard Loss* and the *Average Same Label Loss*, as the first is often used for standard multi-class classification, and the latter may help to perform better at the shared-label prediction task. With that in mind, the idea is to *balance* between these two losses using a hyper-parameter $\lambda$. The balanced loss is defined to be:

$$Balanced\ Loss(batch) = (1 - \lambda) \cdot Average\ Same\ Label\ Loss(batch) + \lambda \cdot Standard\ Loss(batch)$$

Intuitively, the loss function encourages the model to do well on both the $n$-tuple as a whole, and on each specific image (controlled by the hyper-parameter $\lambda$). When $\lambda = 1$, this is equivalent to the baseline method. Throughout our experiments, we use $\lambda = \frac{1}{2}$. For this method, evaluation is done in the same way as in the baseline method.

## 4.3 GRAPH METHOD

In the graph method, we propose a duplicated convolutional neural network architecture with information passage between the different copies of the CNN. This is inspired by recent work on Graphical Neural Networks (GNNs) (Zhou et al., 2018; Kipf & Welling, 2017; Wu et al., 2020). We investigate two kinds of architecture, see Appendix B for further information.

## 4.4 UNIFIED CNN-LSTM METHOD

### 4.4.1 ARCHITECTURE

We propose a unified CNN-LSTM architecture, which effectively learns both the embedding of the data points into low-dimensional vectors and the dependency between the embedding of different data points in the same sequence. An illustration of this architecture is shown in Appendix A.1. The CNN part extracts semantic representations from images, whereas the shared-label dependency between data points in the same sequence in this low-dimensional space is modeled with the long short-term memory (LSTM) recurrent neurons, which maintain the information of label context in their internal memory states (for more information on LSTM, see Appendix A.2). The LSTM part computes the probability of a shared-label prediction sequentially as an ordered prediction path, where the a posteriori probability of the single true label can be computed again at each time step, based on the image embedding at the current time step and the output of the recurrent neuron from the previous time step. The proposed CNN-LSTM model is a unified framework combining the advantages of both learning an effective image embedding using a deep CNN, while also taking into account the label sharing.

### 4.4.2 TRAINING

Training with the unified CNN-LSTM method is done by using the cross entropy loss on the soft-max normalization of the average of the outputs of the linear layer following the LSTM (see Figure 10), and employing back-propagation through time algorithm. Although it is possible to train the model in an end-to-end way, our experiments show that it is much more preferable to train the CNN part separately, using the *balanced method* (Section 4.2), and truncate the final classification layer to achieve the desired embedding. Although it is possible to fine-tune the convolutional neural network afterward, we keep it unchanged in our implementation for simplicity (we noticed that it doesn't make any considerable differences). Both parts of the model are also pruned separately.

When using this method, an important decision is to determine the order of the sequence (as the LSTM part is not symmetric). For further information on the different order techniques that we have experimented with and their motivation, please refer to Appendix A.3.

## 5 EXPERIMENTS

In this section we report the results of our experiments based on the ideas presented in Sections 1.2 and 2. Full experimental details are in Appendix C.

### 5.1 COMPARISON USING THE BASELINE METHOD

In this section, we use the baseline method and a Conv model on the CIFAR-10 data set (see Appendix C for more information). We study how variation in the floating-point operations (FLOPs) due to altering values of $n \in \{1, 2, 3, 4, 5, 7, 10, 15, 40, 60\}$ and $\rho$ effect on different evaluation measures. The results are presented in Figures 1 - 7. Similar results for different combinations of models and data sets are presented in Appendix E.1.

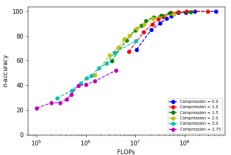
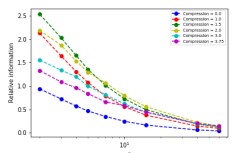
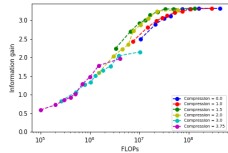

Figure 1: $n$-accuracy and FLOPs comparison. It is observed that different compression ratios are optimal (in terms of FLOPs) for different desired $n$-accuracy.

Figure 2: Log error and FLOPs comparison, where the error is simply $1 - n\text{-accuracy}/100$ and the binary logarithm is used.

Figure 3: Relative information and $n$ comparison. For each value of $n$ we observe a specific order between the compression ratios.

Figure 4: Information gain and FLOPs comparison. It is observed that different compression ratios are optimal (in terms of FLOPs) for different desired information gain.

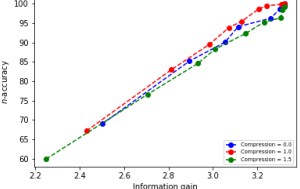
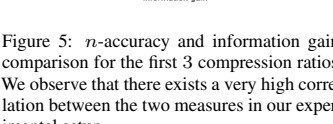
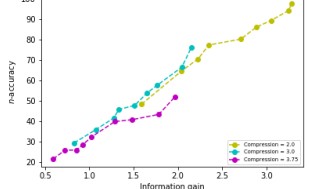
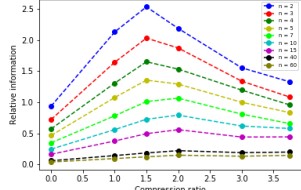

Figure 5: $n$-accuracy and information gain comparison for the first 3 compression ratios. We observe that there exists a very high correlation between the two measures in our experimental setup.

Figure 6: $n$-accuracy and information gain comparison for the last 3 compression ratios. We observe that there exists a very high correlation between the two measures in our experimental setup.

Figure 7: Relative information and compressions comparison for different values of $n$. We observe a *sweet spot* in terms of the compression ratio, for each different value of $n$.

### 5.2 BASELINE AND BALANCED METHODS COMPARISON

In this section, we compare the performances of the baseline method and the balanced method, and report the $n$-accuracy on various data sets, models, and values of $n$ and $\rho$. In all cases, it is

observed that the *balanced* method (with $\lambda = \frac{1}{2}$) outperforms the baseline method in the shared-label prediction task for sufficiently large values of $n$. Nevertheless, it is interesting to observe that the baseline method still almost always outperforms the balanced method in the normal classification task (equivalent to shared-label prediction with $n = 1$). The results are summarized in Table 1 in Appendix F. For further discussion on the comparison between the two methods, please see Appendix D.

### 5.3 SHARED-LABEL PREDICTION METHODS COMPARISON

In this section, we compare the performances of all the shared-label prediction baseline methods discussed above, and report the $n$-accuracy on various data sets, models, and values of $n$ and $\rho$. In all cases, it is observed that the unified CNN-LSTM with *balanced* trained CNN highly outperforms all the other methods for every value of $n \in (2, 5, 7, 15, 40)$ , even though it has less remaining parameters, and uses fewer FLOPs. The results for the Conv model and Tiny-Imagenet data set in measures of $n$-accuracy and relative information are presented in Figure 8 and Figure 9 respectively. A similar figure for the MNIST data set is presented in Appendix E.2. Further results for the higher $n$-accuracy methods, with different combinations of models and data sets are presented in Table 2 in Appendix F.

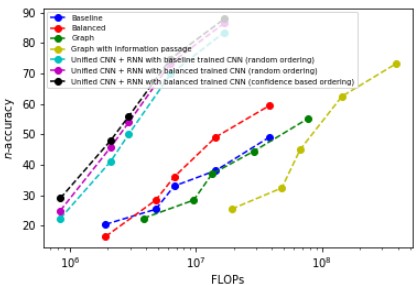

Figure 8: $n$-accuracy on Tiny ImageNet with various shared-label prediction methods.

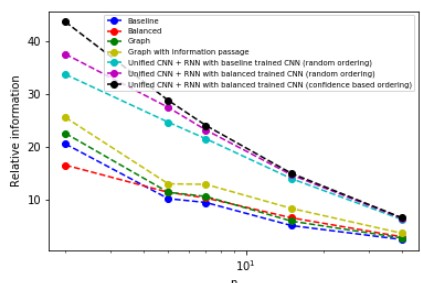

Figure 9: Relative information on Tiny ImageNet with various shared-label prediction methods.

## 6 CONCLUSION AND DISCUSSION

We introduce a real-world motivated problem and investigate how to take advantage of additional side information in order to reduce computational efforts. We study a simple scenario which we coined as the shared-label prediction problem, and suggest various methods, based on different architectures in deep learning, to perform it. We conduct extensive experiments to improve our understanding of i) the and advantages or disadvantage of each method, and the differences between them, ii) the vast connection between measures of accuracy, information, compression ratio, and FLOPs in our settings, and how they interact with each other, and (iii) introduce relative information as a generalized measure of information that the network learns, per computational cost, which, to the best of our knowledge, has not been previously proposed. We further suggest that it enjoys a *sweet spot* phenomenon, that leads to a regime, where in certain scenarios increasing or decreasing the compression ratio knob $\rho$ can deteriorate the relative information. Therefore, we also believe our characterization of the sweet spot provides a useful way of thinking for practitioners.

Throughout our research, we have used common pruning algorithms as a black box. Is it an interesting future research question to ask whether it is possible to design a pruning algorithm (or somehow incorporate an existing one as part of the prediction method) that is better suited for the task of shared-label prediction, namely, one that takes advantage of the side information scenario, in order to gain higher performances.

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

## A    UNIFIED CNN-LSTM

### A.1    ARCHITECTURE ILUSTRATION

The architecture of the proposed unified CNN-LSTM model for shared-label prediction is presented in Figure 10. The convolutional neural network is employed as the image representation, and the recurrent layer captures the information of the previously predicted labels. The outputs of the LSTM are fed through a linear classifier to compute the output label probability.

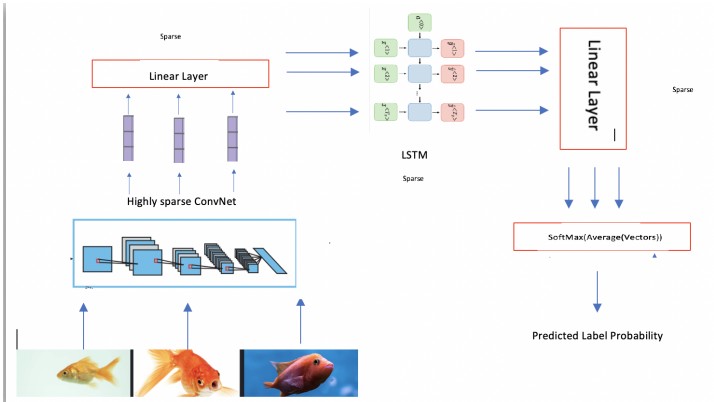

Figure 10: The unified CNN-LSTM architecture

## A.2 Long Short Term Memory Networks (LSTM)

As mentioned earlier, since the objective is to characterize the high-order label dependency in the same sequence (data points embedding in the same sequence share the same label), we employ long short term memory (LSTM) neurons (Hochreiter & Schmidhuber, 1997) as our recurrent neurons. This approach has been demonstrated to be a powerful model of long-term dependency. RNN is a class of neural network that maintains internal hidden states to model the dynamic temporal behavior of sequences with arbitrary lengths through directed cyclic connections between its units. It can be considered as a hidden Markov model extension that employs a nonlinear transition function and is capable of modeling long term temporal dependencies. LSTM extends RNN by adding three gates to an RNN neuron: a forget gate $f$ to control whether to forget the current state; an input gate $i$ to indicate if it should read the input; an output gate $o$ to control whether to output the state. These gates enable LSTM to learn long-term dependency in a sequence, and make it is easier to optimize, because these gates help the input signal to effectively propagate through the recurrent hidden states $r(t)$ without affecting the output. LSTM also effectively deals with the gradient vanishing/exploding issues that commonly appear during RNN training (Pascanu et al., 2012).

## A.3 Sequence order of the LSTM

In the experiments of this paper, we tested both random ordering and confidence based ordering - the sequence order during training (and inference) is determined according to the confidence of the corresponding data points by the CNN model. Data points that have higher confidence in the prediction by the CNN model (trained separately with the balanced method) appear earlier than the less confident ones. This corresponds to the intuition that easier data points should be predicted first to help predict more difficult data points (one data points is classified with higher confidence than the other data point if the largest entry in its logit vector is higher than the largest entry in the other data point logit vector). In particular, the first data point in the sequence will not have a prediction from earlier time to rely on, and we would like this prediction to be as easy as possible. Otherwise, we may face a problem - if the first predicted label is wrong, it is possible that the whole sequence will not be correctly predicted. In our experiments, confidence based ordering usually gained better performances than random ordering, especially for lower values of $n$. We further attempted to randomly permute the label orders in each mini-batch, repeat multiple times, and then taking the average prediction, but this does not have notable effects on the performance and it makes the training harder to converge.

## B The Graph Method

In the graph method, we investigate the following two architectures:

1. Each of the $n$ data points in the $n$-tuple goes through a copy of the CNN, and their $n$ corresponding embeddings are fed through another classifier. The whole architecture is trained end-to-end using the cross entropy loss.

2. Each of the $n$ data points in the $n$-tuple goes through a copy of the CNN, but now, information passes between different copies of each neuron. This is similar to the architecture used in GNN's (Graphical Neural Networks).

## C    EXPERIMENTAL DETAILS

### C.1    MODELS

We use the following architectures as the model/CNN for each method throughout our experiments.

**FC**. Standard fully-connected network designed as follows for input $x$:

$x \leftarrow$ Flatten$[x]$ (Flattens a tensor of dimensions $C \times H \times W$ to a vector of size $C \cdot H \cdot W$)
$x \leftarrow$ ReLU(Linear($C \cdot H \cdot W, 100$))
$x \leftarrow$ ReLU(Linear($100, 100$)$[x]$) (repeat 4 times)
$x \leftarrow$ ReLU(Linear($100, \ell$)$[x]$)

**Conv**.   Standard CNN. We consider a simple 5-layer CNN which is based on the "backbone" architecture from Page (2018), designed as follows for input $x$:

$x \leftarrow$ Conv2d(in channels, out channels $= 32$, kernel size$=(3, 3)$, stride$=(1, 1)$, padding$=(1, 1)$)$[x]$
(in channels $= 1$ for MNIST, in channels $= 3$ for CIFAR10, CIFAR100, Tiny ImageNet)
$x \leftarrow$ ReLU$[x]$
$x \leftarrow$ Conv2d(in channels $= 32$, out channels $= 32$, kernel size$=(3, 3)$, stride$=(1, 1)$, padding$=(1, 1)$)$[x]$
$x \leftarrow$ Flatten(ReLU$[x]$)
$x \leftarrow$ Linear(in features ,$\ell$)$[x]$)

**ResNet18**. (He et al., 2015)

**WideResNet20**. (Zagoruyko & Komodakis, 2016)

### C.2    OTHER EXPERIMENTAL SETUP

**Optimization**. We used the Adam optimizer (Kingma & Ba, 2014), learning rate was set at constant to $10^{-4}$ and all other parameters were set to their default PyTorch values.

**Data sets**. We conducted our experiments on several public benchmark data sets for image classification:

- MNIST (LeCun & Cortes, 2010)
- CIFAR-10 (Krizhevsky, 2009)
- CIFAR-100 (Krizhevsky, 2009)
- Tiny-ImageNet (Tiny-ImageNet)

**Pruning algorithms**.   All pruning algorithms considered here use the following two steps: (i) scoring parameters, and (ii) masking parameters globally across the network with the lowest scores. description of how we compute scores used in each of the pruning algorithms:

- Random: We sampled independently from a standard Gaussian.
- Magnitude: We computed the absolute value of the parameters. (Han et al., 2015)
- SNIP: As done in (Lee et al., 2019)
- GraSP: As done in (Wang et al., 2020)
- SynFlow: As done in (Tanaka et al., 2020)

We report the results using the SynFlow pruning algorithm as it achieved the best results for all methods tested. We run the pruning algorithm for 100 iterations **before** the training phase (our comparisons hold for other pruners as well).

## D    EXTENDED DISCUSSION ON THE BALANCED METHOD

It is further observed from our research that it is possible to improve the results of the balanced method. Higher $n$-accuracy measures were achieved using the following procedure:

- Train a model with the *baseline* method until no further improvement is gained.
- Freeze all the layers in the model, except the last few layers.
- Retrain the mode with the *balanced* method until no further improvement is gained.

The method achieves even higher $n$-accuracy than the baseline method in the shared-label prediction task for sufficiently large values of $n$ , which in turn is better than the baseline method, as reported in Table 1 in Appendix F.

The intuition behind this method is similar to standard transfer learning: Training the model initially with the baseline method yields a model with better representation of both the lower-level features and the higher-level features of the data. Then, tuning it at the end by retraining the final layers only with balanced training yields a model with more adequate high-level features for the shared-label prediction task in one hand, and a better understanding of how to use these features to make a better decision, based on various data points containing the same label.

Consider the following example when classifying MNIST digits: A standard model in the highly-parametrized regime would learn to detect basic features such as small curved lines in the shallow part of the neural network, and at a deeper stage of the network it may learn to detect more complex features such as circles. When we are at the low-parametrized regime, we have to make a compromise, deteriorating the quality of the high-level features. Using this method may benefit the model by helping it to learn better "low quality" high-level features that may not be "good enough" for normal classification, yet are sufficient for the shared-label prediction task.

## E    ADDITIONAL PLOTS

### E.1    PLOTS FROM SECTION 5.1

Figures 11-20 describe the results of the comparison done as part of the experiments in Section 5.1. The results presented in Figures 11-15 were generated using a Wide-ResNet20 model on the Tiny-ImageNet data set ($n \in \{1, 2, 3, 4, 5, 7, 10, 15, 40\}$), and the results presented in Figures 16-20 were generated using a ResNet18 model on the CIFAR-100 data set ($n \in \{1, 2, 3, 4, 5, 7, 10, 15, 40, 60\}$).

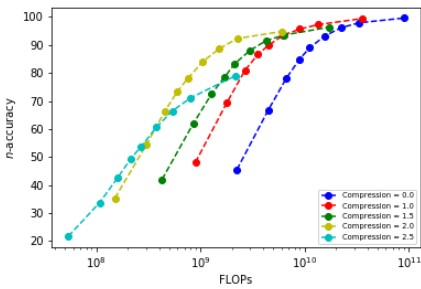

Figure 11: $n$-accuracy and FLOPs comparison. It is observed that different compression ratios are optimal (in terms of FLOPs) for different desired $n$-accuracy.

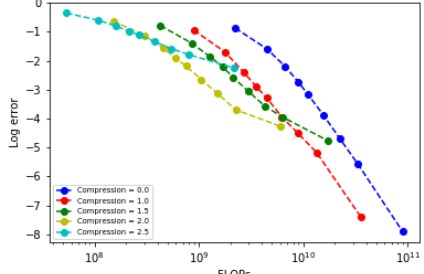

Figure 12: Log error and FLOPs comparison, where the error is simply $1 - n$-accuracy$/100$ and the binary logarithm is used.

### E.2    PLOTS FROM SECTION 5.3

Figures 21-22 describe the results of the comparison done as part of the experiments in Section 5.3. The results presented in figure were generated using a Conv model on the MNIST data set ($n \in \{1, 2, 3, 4, 5, 7, 10, 15, 40, 60\}$).

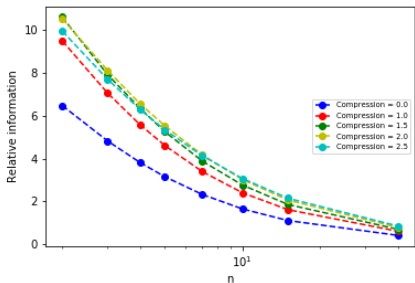

Figure 13: Relative information and $n$ comparison. For each value of $n$ we observe a specific order between the compression ratios.

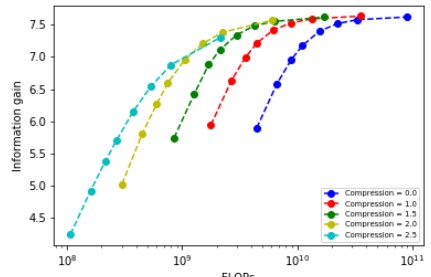

Figure 14: Information gain and FLOPs comparison. It is observed that different compression ratios are optimal (in terms of FLOPs) for different desired information gain.

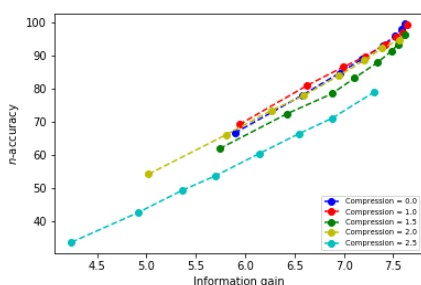

Figure 15: $n$-accuracy and information gain comparison. We observe that there exists a very high correlation between the two measures in our experimental setup.

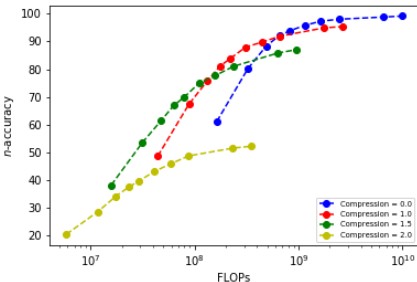

Figure 16: $n$-accuracy and FLOPs comparison. It is observed that different compression ratios are optimal (in terms of FLOPs) for different desired $n$-accuracy.

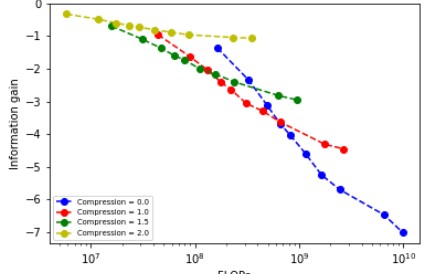

Figure 17: Log error and FLOPs comparison, where the error is simply $1 - n\text{-accuracy}/100$ and the binary logarithm is used.

# F   TABLES

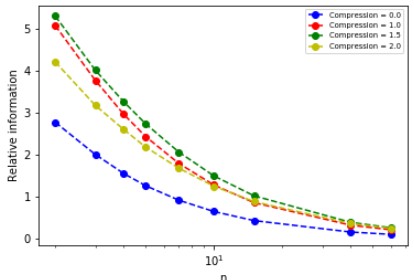

Figure 18: Relative information and $n$ comparison. For each value of $n$ we observe a specific order between the compression ratios.

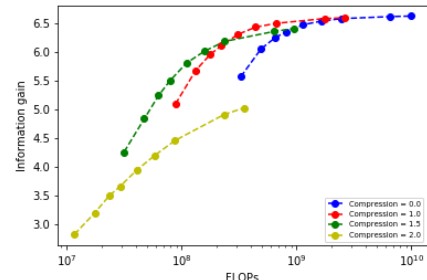

Figure 19: Information gain and FLOPs comparison. It is observed that different compression ratios are optimal (in terms of FLOPs) for different desired information gain.

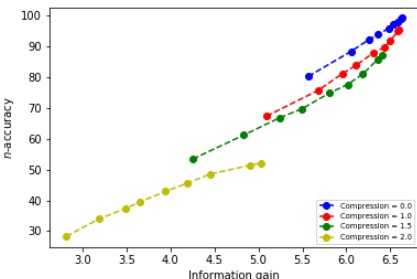

Figure 20: $n$-accuracy and information gain comparison. We observe that there exists a very high correlation between the two measures in our experimental setup.

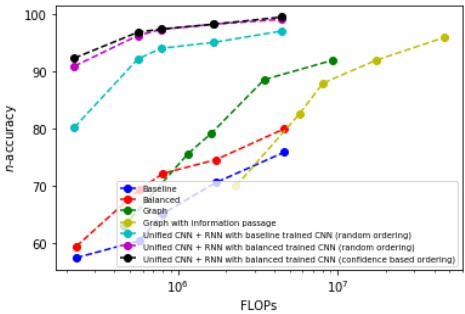

Figure 21: $n$-accuracy on MNIST with various shared-label prediction methods.

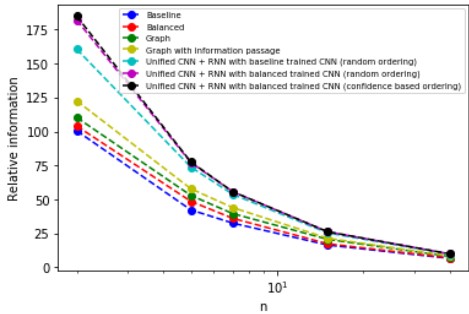

Figure 22: Relative information on MNIST with various shared-label prediction methods.

| Method | Data set | Total parameters | Compression | Remaining parameters | Model | $n=1$ | $n=25$ | $n=40$ | $n=60$ | $n=80$ |
|---|---|---|---|---|---|---|---|---|---|---|
| Baseline | MNIST | 119,400 | 2.5 | 378 | FC | **35.20** | 71.72 | 74.69 | 78.26 | 79.17 |
| Balanced | | | | | | 30.29 | **78.28** | **86.53** | **89.44** | **93.33** |
| Baseline | MNIST | 260,384 | 3.25 | 147 | Conv | **55.41** | 92.68 | 95.92 | 97.52 | 96.67 |
| Balanced | | | | | | 50.69 | **94.44** | **98.78** | **100** | **100** |
| Baseline | MNIST | 260,384 | 3.5 | 83 | Conv | **39.20** | 75.51 | 80.00 | 81.99 | 86.67 |
| Balanced | | | | | | 35.15 | **78.79** | **85.31** | **90.06** | **94.17** |
| Baseline | CIFAR-10 | 337,760 | 2.0 | 3,378 | Conv | **50.54** | 91.75 | 90.40 | 91.25 | 91.67 |
| Balanced | | | | | | 35.16 | **92.25** | **96.80** | **99.38** | **100** |
| Baseline | CIFAR-10 | 11,172,032 | 4.5 | 354 | ResNet-18 | **28.92** | 45.25 | 44.00 | 41.88 | 40.00 |
| Balanced | | | | | | 21.10 | **50.75** | **60.40** | **67.50** | **75.83** |
| Baseline | CIFAR-100 | 3,286,880 | 2.5 | 10,395 | Conv | 18.28 | 77.00 | 78.50 | 78.00 | 83.00 |
| Balanced | | | | | | **18.35** | **80.00** | **83.00** | **85.00** | **88.00** |
| Baseline | CIFAR-100 | 1,092,960 | 2.0 | 10,930 | WideResNet-20 | **21.10** | 59.75 | 61.50 | 61.00 | 62.00 |
| Balanced | | | | | | 15.55 | **63.75** | **71.20** | **81.00** | **88.00** |

Table 1: $n$-accuracy on different data sets with various architectures and compressions. In all cases, the balanced method highly outperforms the balanced method (for sufficiently large values of $n$).

| Method | Data set | Total parameters | Compression | Remaining parameters | Model | n=2 | n=5 | n=7 | n=15 | n=40 |
|---|---|---|---|---|---|---|---|---|---|---|
| Balanced | | 260,384 | | 83 | Conv | | | | | |
| Unified CNN-LSTM with baseline trained CNN | MNIST | 260,384 + 171,710 = 432,094 | 3.5 / 3.75, 3.75 | 47 + 31 = **78** | Conv + LSTM | 59.43 / 80.25 | 69.44 / 92.24 | 72.70 / 94.10 | 74.61 / 95.15 | 85.31 / 97.14 |
| Unified CNN-LSTM with balanced trained CNN | | 260,384 + 171,710 = 432,094 | 3.75, 3.75 | 47 + 31 = **78** | Conv + LSTM | **90.97** | **96.34** | **97.40** | **98.33** | **99.18** |
| Balanced | | 11,172,032 | | 1118 | ResNet-18 | | | | | |
| Unified CNN-LSTM with baseline trained CNN | CIFAR-10 | 11,172,032 + 176,730 = 11,348,762 | 4.0 / 4.5, 2.5 | 354 + 559 = **913** | ResNet-18 + LSTM | 58.10 / 60.10 | 73.05 / 79.40 | 77.96 / 83.59 | 84.39 / 89.09 | 89.60 / 93.60 |
| Unified CNN-LSTM with balanced trained CNN | | 11,172,032 + 176,730 = 11,348,762 | 4.5, 2.5 | 354 + 559 = **913** | ResNet-18 + LSTM | **62.06** | **81.15** | **85.42** | **94.24** | **98.40** |
| Balanced | | 1,092,960 | | 10,930 | WideResNet-20 | | | | | |
| Unified CNN-LSTM with baseline trained CNN | CIFAR-100 | 1,092,960 + 312,900 = 1,405,860 | 2.0 / 2.5, 2.0 | 3457 + 3129 = **6586** | WideResNet-20 + LSTM | 33.34 / 39.86 | 48.05 / 60.95 | 51.43 / 70.64 | 58.83 / 80.16 | 71.20 / 88.50 |
| Unified CNN-LSTM with balanced trained CNN | | 1,092,960 + 312,900 = 1,405,860 | 2.5, 2.0 | 3457 + 3129 = **6586** | WideResNet-20 + LSTM | **42.5** | **64.70** | **73.07** | **84.16** | **92.00** |
| Balanced | | 26,224,480 | | 262,245 | Conv | | | | | |
| Unified CNN-LSTM with baseline trained CNN | Tiny ImageNet | 26,224,480 + 401,800 = 26,626,280 | 2.0 / 3.0, 1.0 | 26,225 + 40,180 = **66,405** | Conv + LSTM | 16.52 / 22.44 | 28.45 / 41.05 | 36.07 / 50.14 | 49.17 / 69.83 | 59.50 / 83.50 |
| Unified CNN-LSTM with balanced trained CNN | | 26,224,480 + 401,800 = 26,626,280 | 3.0, 1.0 | 26,225 + 40,180 = **66,405** | Conv + LSTM | **25.0** | **45.70** | **54.00** | **73.33** | **86.50** |

Table 2: $n$-accuracy on different data sets with various architectures and compressions. In all cases, the unified CNN-LSTM with balanced trained CNN highly outperforms the other methods for every value of $n$, even though it has less remaining parameters and uses less FLOPs.

