# OpenReview forum: "More Side Information, Better Pruning: Shared-Label Classification as a Case Study"
_ICLR.cc/2021/Conference — Reject_

### Official Review · AnonReviewer4 · 2020-10-26
**Lacking clarity**

**Rating:** 4
**Confidence:** 3

**Review:**

Summary

This paper introduces the problem of shared-label prediction -- the problem of classifying the (common) label of a set of points conditioned on the knowledge that they all share the same label --  and suggests various methods that take advantage of side information to solve it.

Rationale for Score

I think that the idea of using side information for pruning is promising overall. However, given the lack of clarity in the exposition -- e.g., in motivating and defining the problem (and its connections to pruning) and conveying & contextualizing the paper's contributions --, I am unable to fully understand and appreciate the significance of this work.

Strengths

- The shared-label prediction problem introduced seems interesting and made me initially wonder whether it could be applied to, e.g., accelerating pooled-testing methods for Covid-19 (with neural nets) where a small partitioned group of samples may share the label (negative) with high probability
- Using side information for pruning is a nice idea that may be appreciated by the pruning community
- There are some empirical evaluations of the proposed approaches


Weaknesses

- The abstract is very long, takes up almost the entire first page, and reads more like an introduction than a proper abstract. Despite its length, the proposed method or novel idea of the paper is not revealed, but rather only merely hinted at as “we propose various methods for performing this task.”
- The problem that this paper attempts to solve is not very well motivated by a concrete real-world application as the abstract suggests. As someone that is not very familiar with the problem tackled by the paper, I am left trying to think of a scenario where we *know for a fact* a priori that the group of points we are feeding as input to the neural network share the same label, but yet do not know the label itself.
- The problem definition (and consequently, my sense of the paper’s contribution) is very confusing. There are numerous “problems of interest” introduced in the paper right off the bat and it is very difficult to discern the particular problem that serves as the main focus of the paper. In particular, the paper starts off in Sec. 1.1 with a very generic problem of trying to maximize the ground-truth “retrieval function given a static space of objects.” This is then relaxed to using approximations of the ground-truth retrieval function in terms of a neural network, which is then reformulated as the problem of using a pruned network to construct a shortlist of candidates for which the original network is used. Then, the authors claim that elements of this shortlist have similarity that we can exploit (for reasons that are not clear to me); unfortunately, immediately after the problem is then recast again as one of classification where the input space is partitioned into k clusters, each consisting of points of the same label, and the objective is to classify the label of each cluster using n random samples using the information that all points in the cluster share a label. This is then somewhat more rigorously defined as the “shared prediction problem” in Sec. 1.2 and set as the target problem that this paper tackles.

 Overall, I found this exposition quite confusing and not very well-introduced or motivated by a real-world application (as the authors had hinted at in the abstract). Since the problem in question is quite general, I am also not sure why the pivot of the paper is to emphasize pruning in the introduction, rather than as a potential application of the proposed approach in, e.g., the experiments section. Since the shared prediction problem is more concrete relative to the more general problem that the authors start off with in Sec. 1.1, I would recommend simply defining the shared prediction problem (the most concrete of them all) first, rather than starting with the most abstract problem.
- The definitions for the variables used are ambiguous and defined way later after being used. For example, the compression ratio \rho, which is used as early as Sec. 2, is not formally defined until Sec. 4. This might have been fine if the definition of the compression ratio was consistent with that of existing work -- e.g., (# of parameters in the original network) / (# of parameters remaining in pruned network) -- however, it turns out that, as defined much later on in Sec. 4, it is defined as “logarithm to the base 10 of the number of parameters in the original network divided by the number of parameters remaining after pruning.” This definition is quite confusing and misplaced under the section “Our Methods.” For clarity, I would recommend defining the variable earlier on when it is used in Sec. 2 (to define the problem), and using actual math to define the compression ratio, among other pertinent variables.

---

### Official Review · AnonReviewer1 · 2020-10-26
**The paper discusses empirically an interesting network pruning workflow with side information provided by data instances sharing the same class labels.**

**Rating:** 6
**Confidence:** 3

**Review:**

This paper discusses empirically a workflow about how to compress network architectures with side information. The side information defined in this study is provied by the training data instances that share the same class labels, aka the problem of shared label prediction.

The contribution of this work can be concluded as
1. This work defines a set of benchmarks measuring efficiency of shared label prediction, including relative information and information gain. These two metrics are used to measure how much information can be learned by the compressed neural networks over the data instances sharing the same label, given the compression ratio and the name of training instances
2. This work empirically unveil the sweet spot phenomena, which indicates how relative information varies in an montonical way with respect to different compression ratios.
3. This work proposes to make full use of the training instances sharing the same class label via a combination of CNN and LSTM. CNN is used to extract features and LSTM is used to encode the correlation between different instances sharing the label. Experimental study confirms the merit of the proposed workflow.

Overall, this paper is well explicated, starting with clearly written background on basic concepts and prior work, stating clear the algorithmic design and conducting correspondingly the experimental study to confirm the benefits of the algorithm.

There are several downsides:

1) no theoretical discussion about the sweet spot phenomena is given. The turing point shown in this observation is very interesting. If we can provide an estimate about when the turining point appear (given a compression ratio and a base neural network architecure), that would be very useful for guding practical network compression tasks.

2) Can we still observe the sweet sopt thing for the proposed CNN + RNN workflow ? It looks like Figure 8 and 9 both assume the compression ration is fixed.

3) For the RNN architecture, why does the confidence ordering based RNN perform better than random ordering?

4) When n =1,  it is not surprising to find the balance method has a worse performance than the baseline method. When n = 1, The average batch prediction includes a blurred prediction result by taking the average of k training instances (when n = 1, each n-tuple contains only 1 instance). Thus the average same label loss should be noisy.

---

### Official Review · AnonReviewer2 · 2020-10-30
**Very confusingly written paper**

**Rating:** 2
**Confidence:** 4

**Review:**

The paper proposes to use a set of input examples, x1 to xn, having a common label y, and use them together for better classification. And use these shared label examples as additional information during model pruning.

I have multiple challenges to understand this research paper:

1. Clarity of Writing:
The paper is very tough to read and understand. The authors jump through multiple levels of motivations, starting with information retrieval and then to approximation of database queries. But then the rest of the paper talks about loss averaging and results are shown using a CNN model on CIFAR10, and TinyImagenet. Either the whole motivation on IR aspects can be removed or relevant experiments and approach be proposed. At multiple places, I am either lost or confused on what is the problem that the paper is trying to solve. Example, after reading 2 pages of the paper, the authors state, "We depart from the original motivating information retrieval scenario, and henceforth consider a simpler, toy problem which we call the shared-label prediction problem."

2. Misleading Title and Takeaway in the paper:
The paper title, abstract, and motivation says to use "More Side Information". While shared-labels is not side information or additional information. If you have 10 instances to classify, instead of classifying them independently, the paper is trying to classify them together. So, there is no side information used in the approach. Also, this is not a "structure" present in X.

3. Incorrect or Insufficient assumptions:
The paper makes a lot of strong assumptions, which are not practical:
a. What if multiple instance, x1 to xn of the same class label is not available ? In few shot learning or one shot learning scenario.
b. "We assume the network and the pruning methods are state-of-the-art, and it is not our goal here to improve them" -  I do not understand the need for introduction model pruning for shared label classification. If the goal is not to improve pruning methods, then why do network pruning, at all?
c. "For all methods we study, fixing the information size n, our experiments suggest that there exists a sweet spot phenomenon, or a "compression threshold" in the sense that RI, as a function of ρ, has a global maximum ρ∗" - There is no proven approach that such a threshold should exist for any dataset/model combination. I do not agree to this assumption.

4. Lack of Novelty:
Most of the approaches explained in Section 4 are just averaging the loss of the tuple of samples. When we average the samples, it is automatically considered that the tuple of samples are drawn from i.i.d. And in the proposed CNN-LSTM approach, the tuple of samples are "sequentially" classified. That raises more questions than answers - why sequence, and in what sequence? While the paper does not discuss any of these important questions.

5. Appendix has more information than the paper itself:
Most of the detailed and important information about the paper, including primary details about the approach, the model architecture, and many experimental results are in the appendix. At many instances, the paper reads like an index to the appendix. Example, section 4.3, "We investigate two kinds of architecture, see Appendix B for further information". Throughout the paper we do not have information the architectures. Even the proposed approach in 4.4, "An illustration of this architecture is shown in Appendix A.1"

6. Lack of experiments:
The proposed approach of CNN-LSTM is comapred with baseline methods : loss averaging, graph based averaging of loss. And the paper claims that the CNN-LSTM approach is better than the baseline methods. This is insufficient. The paper fails to compare with other relevant techniques in literature and place the paper empirically among the others papers in the literature.

---

### Official Review · AnonReviewer3 · 2020-11-01
**Rigorous experimental results, but insufficient motivation for proposed problem**

**Rating:** 4
**Confidence:** 4

**Review:**

More Side Information, Better Pruning: Shared-Label Classification as a Case Study

Summary:
The goal of this paper is to use side information about a task to prune models more effectively i.e., with minimal loss in performance as compared to original model.
The particular type of side information they focus on is prior knowledge about a collection of instances sharing a class label. The motivation is an Information Retrieval Scenario, wherein it is expensive to identify relevant examples for a query, therefore, an approximate, cheaper model is used to identify good candidates that have a high likelihood of being relevant. The original model is then computed only on the identified subset. The paper then switches to using a related, toy problem where the goal is to predict an unknown, shared label for a given tuple of n items.
The paper describes 4 methods to exploit the additional information: (i) a baseline method that trains in a standard fashion and computes the label of each example in an n-tuple independently. (ii) Balanced method: Sum of standard classification loss and cross entropy computed with respect to average labels per batch
(iii) Graph method based on GNNs (iv) CNN+LSTM architecture, where n-tuples with shared labels are treated as a sequence passed to an LSTM.
Finally, the paper proposes a relative information metric that measures the tradeoff of information gain vs the computation cost. Empirical results are presented comparing the various methods, showing the relationship between relative information, compression ratios and n.

Strengths:
1. Authors propose a novel quantitative metric (relative information gain) in measuring the loss of performance in pruned models vs computational cost. This gives practioners a tool to clearly think about tradeoffs in cost vs model certainty.
2. Paper provides rigorous experimental results. Each proposed method is compared under various compression ratios. Relative information is shown to be correlated with both compression ratio and n (number of examples in a tuple with shared label). For a given compression ratio, relative information is shown to improve with increasing n until a "sweet spot" is reached, beyond which relative information starts degrading.
Relationship between accuracy at various compression ratios and FLOPs is also reported clearly.

Weaknesses:
1. The paper is missing a clear description of real-world applications.
a. The original motivation is a very interesting problem, wherein only an approximate function can be computed generating a subset that *possibly* share the same label. However, the actual task the rest of the paper focuses on is materially different wherein n examples are known to share the same label. Can the authors describe a real-world scenario where one is guaranteed to receive n examples at a time belonging to a single class.
b. If these n examples come from an approximate classifier as in the original motivating scenario, how do the methods described in this paper handle "within-tuple" uncertainty, i.e. uncertainty that all the examples indeed belonging to the same class. If we had reasonable certainty in all examples having the same label, then why do we need another more complex/expensive classifier to be applied subsequently?
2. The methods are not clearly described.
a. For the balanced method, a portion of the loss per batch is cross-entropy between average label of an n-tuple and the true label of the n-tuple, averaged across k n-tuples in the batch. This is not clearly described at all and required many readings. Notation is unclear: both $k$ and $l$ are used to describe the number of n-tuples in a batch. k is also used to describe the number of data points belonging to a single label i. In that case, different labels $i$ have different sizes $k_i$. After k is introduced, it is not used at all.
b. Graph method is described in 1 sentence. Some information about the architecture is in the appendix which are not very helpful either. Paper needs to be self-contained.  What is the input graph passed to the GNN? Assuming, all examples with the same label are represented as fully connected subgraphs, does the complete dataset comprise of several disconnected components. Is the performance impacted by number of subgraphs/labels?
c. In the LSTM based method, an n-tuple is treated as a sequence, so that LSTMs can used to capture the fact that the examples in an n-tuple are related. This is unusual usage, since LSTMs would only be able to model local neighborhood in practice. Paper claims that ordering examples by certainty gives improved performance, and supports this claim with empirical results. It is unclear, how well the models cope with errors in confidence or uncalibrated models. The current set of experiments do not address this key factor that would occur in any practical setting.

Conclusion:
My recommendation is to reject the paper at this time, because the problem statement is not well-formed. Specifically, how the methods handle uncertainty of labels within a tuple. This is especially confusing given that LSTM-based solution is found to be the best empirically. However, how would an LSTM perform if intermediate examples are in fact mislabeled. Additionally, the explanation of the methods is unclear.

I would encourage the authors to make their future work stronger by grounding the work in a real problem. The IR example cited at the beginning is a good one. If this problem is solved as is, this work can be very impactful. The simplifying assumptions made at the moment weaken the problem statement. Therefore, even though the experimental results are thorough, their application to any practical scenario is not obvious.

---

### Official Review · AnonReviewer5 · 2020-11-06
**Recommendation to Reject**

**Rating:** 3
**Confidence:** 4

**Review:**

The authors study how to improve the prediction and pruning performance with additional information generated by labels in the shared-label classification problem. As a starting point, the authors consider a simple scenario where side information can be extracted from the same labeled batch. To train the neural network, the authors use a balanced loss consisting of a weighted sum of general cross-entropy and cross-entropy of average batch prediction. The authors also suggest a new CNN-LSTM architecture to improve predictive performance to exploit the side information. The experiments section shows the proposed method performs well and achieves a high compression rate.

The data model used in this study is different from the common classification problem. This paper assumes that n-data points give side information with the shared-label batch, referred to as "n-tuple." In general, classification problems have labels independent of other data points.

[Strength]

The authors study a general relationship between pruning and additional side information for the shared label problem. The authors define "relative information" to measure the appropriate compression rate for various prediction performance and pruning levels. Using the relative information, we can dramatically reduce the number of parameters while maintaining prediction performance. Besides, the proposed CNN-LSTM architecture improves the prediction performance with the shared-label training scheme.

[Weakness]

This paper considers a very different data model which never been studied before. The shared-label model should be motivated very well. I'm not convinced why we have to study this model.

The explanation of which benefits the network from the side information is ambiguous.  There is no theoretical and empirical explanation of how the side information and balanced loss can discriminate ineffective parameters. It requires providing more clear evidence, such as the statistics of network parameters before and after the pruning algorithm.

The existence of optimal compression rate \rho^* should be discussed more rigorously if possible with a theoretical proof based on relative information.

In the experiment section, only the 5 Conv net was investigated to check the effectiveness of pruning. It would be more convincing if the authors can add results for the proposed CNN-LSTM network.

[Minor Issue]

It is challenging to read Fig1-9 because of their interpretation. I suggest the authors use other colors with a bigger font size.

It would be better to use tables rather than graphs to present many experimental results. Graphs are too small to understand

---

### Decision · Program_Chairs · 2021-01-07
**Final Decision**

**Decision:**

Reject

**Comment:**

This paper tackles the problem of classifying a set of points given the knowledge that all points should have the same class. There seems to be a consensus among the reviews that under the assumptions made, the authors provide thorough experiments convincing that their method is useful. However, the paper has two weaknesses that are too strong to ignore. First, the clarity of exposition seems to be lacking, specifically a clear motivation for this new setup as well as its connection to the abstract problem being solved. Second, the assumptions made seem to be too strong, and the solution seems to rely on these strong assumptions too much.